# Multidrug-Resistant *Staphylococcus* sp. and *Enterococcus* sp. in Municipal and Hospital Wastewater: A Longitudinal Study

**DOI:** 10.3390/microorganisms12040645

**Published:** 2024-03-24

**Authors:** Maria Elena Velazquez-Meza, Miguel Galarde-López, Patricia Cornejo-Juárez, Berta Alicia Carrillo-Quiroz, Consuelo Velázquez-Acosta, Miriam Bobadilla-del-Valle, Alfredo Ponce-de-León, Celia Mercedes Alpuche-Aranda

**Affiliations:** 1Centro de Investigación Sobre Enfermedades Infecciosas, Instituto Nacional de Salud Pública, Cuernavaca City 62100, Mexico; mevelaz@insp.mx (M.E.V.-M.); miguel.galarde@insp.edu.mx (M.G.-L.); berta.carrillo@insp.mx (B.A.C.-Q.); 2Departamento de Infectología, Instituto Nacional de Cancerología, Mexico City 14080, Mexico; patcornejo@yahoo.com (P.C.-J.); consueve62@yahoo.com.mx (C.V.-A.); 3Laboratorio Nacional de Máxima Seguridad para el Estudio de Tuberculosis y Enfermedades Emergentes, Instituto Nacional de Ciencias Médicas y Nutrición “Salvador Zubirán”, Mexico City 14080, Mexico; mbv99@hotmail.com (M.B.-d.-V.); alf.poncedeleon@gmail.com (A.P.-d.-L.)

**Keywords:** wastewater, multidrug resistance, persistence, *Staphylococcus* sp., *Enterococcus* sp.

## Abstract

The objective of the study was to detect multidrug-resistant *Staphylococcus* sp. and *Enterococcus* sp. isolates in municipal and hospital wastewater and to determine their elimination or persistence after wastewater treatment. Between August 2021 and September 2022, raw and treated wastewater samples were collected at two hospital and two community wastewater treatment plants (WWTPs). In each season of the year, two treated and two raw wastewater samples were collected in duplicate at each of the WWTPs studied. Screening and presumptive identification of staphylococci and enterococci was performed using chromoagars, and identification was performed with the Matrix Assisted Laser Desorption Ionization Time of Flight mass spectrometry (MALDI-TOF MS^®^). Antimicrobial susceptibility was performed using VITEK 2^®^ automated system. There were 56 wastewater samples obtained during the study period. A total of 182 *Staphylococcus* sp. and 248 *Enterococcus* sp. were identified. The highest frequency of *Staphylococcus* sp. isolation was in spring and summer (n = 129, 70.8%), and for *Enterococcus* sp. it was in autumn and winter (n = 143, 57.7%). Sixteen isolates of *Staphylococcus* sp. and sixty-three of *Enterococcus* sp. persisted during WWTP treatments. Thirteen species of staphylococci and seven species of enterococci were identified. Thirty-one isolates of *Staphylococcus* sp. and ninety-four of *Enterococcus* sp. were multidrug-resistant. Resistance to vancomycin (1.1%), linezolid (2.7%), and daptomycin (8.2%/10.9%%), and a lower susceptibility to tigecycline (2.7%), was observed. This study evidences the presence of *Staphylococcus* sp. and *Enterococcus* sp. resistant to antibiotics of last choice of clinical treatment, in community and hospital wastewater and their ability to survive WWTP treatment systems.

## 1. Introduction

Antimicrobial resistance (AMR) is a global public health problem associated with increased healthcare costs, prolonged hospitalizations, morbidity, and mortality [1]. It is estimated that by 2050, 300 million people will die from AMR if this problem is not overcome. This will result in a loss of economic output of between USD 60 billion and USD 100 billion. AMR will have severe adverse effects on the global economy. If not addressed now, the world’s gross production may decrease by USD 8 trillion per year by 2050 [2]. Multiple factors contribute to the emergence of multidrug resistant bacteria, including inappropriate use of antimicrobials, large amounts of antibiotics used for human and animal therapy, and prolonged therapies, among others. Multidrug resistance in bacteria is caused by the accumulation of resistance genes in the chromosome or in mobile genetic elements which encode resistance to specific antimicrobials, and by the action of efflux pumps that expel more than one type of antimicrobial. All this has led to the emergence of multidrug-resistant bacteria, called “superbugs” because they cannot be treated with most existing antimicrobials [3]. This will pose significant economic challenges to healthcare systems in effectively managing infections caused by these bacteria. It is estimated that the increased cost of treating a patient with a resistant pathogen can be as high as 30% [3,4]. Approaches to combating AMR include humans, animals, plants, and the environment. Among the studies of AMR in the environment, wastewater is of the greatest interest, gathering, because hospital or community wastewater is considered a hotspot or reservoir for antibiotic-resistant bacteria (ARB) and antibiotic-resistant genes (ARG) [5,6,7,8].

Within this group of ARBs are *Staphylococcus* sp. and *Enterococcus* sp. *Staphylococcus* sp. is normal flora found in mucus membranes and the skin of human and other mammals. There are currently more than 40 species of coagulase-negative staphylococci (CoNS), with *Staphylococcus epidermidis*, *Staphylococcus saprophyticus*, and *Staphylococcus haemolyticus* being the main causes of infections in humans [9]. Other species, such as *Staphylococcus hominis*, *Staphylococcus warneri*, *Staphylococcus simulans*, *Staphylococcus saccharolyticus*, *Staphylococcus capitis*, *Staphylococcus cohnii*, *Staphylococcus lugdunensis*, and *Staphylococcus xylosus*, have been isolated from various infections [10]. Coagulase-positive *Staphylococcus aureus* is another species associated with serious infections in humans and animals [11].

Enterococci are Gram-positive bacteria, commensals in the intestinal system of numerous animals, including humans. The two most common species of enterococci are *Enterococcus faecalis* and *Enterococcus faecium*, which are associated with various infections of the urinary tract, surgical wounds, and bloodstream. They are also indicators of fecal contamination [12,13]. These bacteria cause mainly hospital but also community infections that can be difficult to treat due to their resistance to antibiotics, primarily vancomycin, linezolid, daptomycin, and tigecycline, which are considered antibiotics of last choice [14]. These bacteria and their resistance genes can reach the environment through hospital and community wastewater discharges [15,16,17,18]. The role of wastewater in the acquisition and dissemination of AMR is currently a topic of interest which has been addressed in other parts of the world [5,6,7,8], but in our country this evidence is almost nil. The study of these ecosystems can serve as a model for the early surveillance of AMR and for the formulation of proposals for the regulation of discharges of multidrug-resistant bacteria. The objective of the study was to detect multidrug-resistant *Staphylococcus* sp. and *Enterococcus* sp. isolates in municipal and hospital wastewater and to determine their elimination or persistence after wastewater treatment.

## 2. Materials and Methods

### 2.1. Study Sites and Sample Collection

The study was conducted in Mexico City and Cuernavaca City from August 2021 to September 2022. Samples of raw (100 mL) and treated (200 mL) wastewater were collected in each season of the year; two samples of treated wastewater and two samples of raw wastewater in each of the wastewater treatment plants (WWTPs) were studied. Samples were taken at one-month intervals between each sample. The WWTPs studied were the community WWTPs Acapantzingo (ACA) in Cuernavaca City and Coyoacán (COY) in Mexico City, a hospital WWTP (CAN), as well as raw wastewater samples from a second hospital with a single sump (NUT), both located in Mexico City.

The hospital WWTPs included in this study were selected because they are tertiary-level hospitals that serve a large population in Mexico City and surrounding cities, as well as for their location and ease of sampling. Mexico City has 22,281,442 inhabitants, making it one of the most populated cities in Mexico. The community WWTPs were selected because they are among the largest in Cuernavaca City and Mexico City, and they collect wastewater from open population and hospital areas, as well as for their ease of sampling. The WWTPs sampled in this work have primary, secondary, and tertiary treatment systems. The layout of both WWTPs begins with a primary treatment, and then the water flow is dosed to an aeration process mediated by regulating valves in different chambers. The treatment process continues with a secondary phase of clarification and composite sedimentation, precipitating the sludge. The tertiary treatment includes calcium hypochlorite tablets, granular activated carbon filters, and ultraviolet light (UVL) in the effluent. One liter of each wastewater sample was collected using the simple grab technique in sterile containers, and then transported to the to the Research Center of Infectious Diseases at the National Institute of Public Health at 4 °C in less than two hours of collection [19].

### 2.2. Microbial Analysis and Identification

Microbial culturing was performed within three hours of sample collection. The raw wastewater and treated wastewater samples were centrifuged at 5000 rpm for 20 min; the pellet was dissolved in 10 mL of phosphate buffer solution (PBS). Before plating, the samples were diluted 1:1000 in PBS. Then they were plated in Hi-Crome Rapid Hi-Enterococci Agar™, Hi-Crome *Enterococcus faecium* Agar Base™, Hi-Crome Staph Agar Base™, (HiMedia Laboratories, Kennett Square, PA, USA) Trypticase Soy Agar^®^ (MSD, Rahway, NJ, USA) and Mannitol Salt Agar^®^ (MSD, Rahway, NJ, USA) at 37 °C for 18 h. The different suspected staphylococcal and enterococcal colonies isolated from the raw wastewater and treated wastewater were identified first with mass spectrometry using the Matrix Assisted Laser Desorption Ionization Time of Flight mass spectrometry (MALDI-TOF MS^®^) (Bruker Daltonics, Bremen, Germany) equipment.

### 2.3. Antimicrobial Susceptibility Testing

The antimicrobial susceptibility profiles of *Staphylococcus* sp. (n = 182) [100 community wastewater/82 Hospital wastewater] and *Enterococcus* sp. (n = 248) [95 community wastewater/153 Hospital wastewater] isolates were performed using the VITEK 2® automated system (BioMérieux, Marcy l’Etoile, France). A panel of 18 antimicrobials was used: Ampicillin (AMP), cefoxitin (FOX), oxacillin (OXA), gentamicin (GEN), ciprofloxacin (CIP), levofloxacin (LVX), moxifloxacin (MFX), erythromycin (ERY), clindamycin (CLI), linezolid (LZD), daptomycin (DAP), vancomycin (VAN), doxycycline (DOX), tetracycline (TET), tigecycline (TIG), nitrofurantoin (NIT), rifampicin (RIF), and trimethoprim/sulfamethoxazole (TMP/SXT). The minimum inhibitory concentration (MIC) was interpreted according to CLSI, 2022 document [20].

## 3. Results

### 3.1. Staphylococcus sp. and Enterococcus sp. Isolates

There were 56 wastewater samples obtained during the study period. From these, 182 *Staphylococcus* sp. were identified: ACA (n = 65), COY (n = 35), CAN (n = 44), and NUT (n = 38). Community and hospital WWTPs had a similar number of isolates, n = 100 and n = 82, respectively. The seasons with the highest frequency of *Staphylococcus* sp. isolates were spring and summer (n = 129, 70.8%) (Figure 1).

In raw wastewater 166 (91.2%) isolates of *Staphylococcus* sp. were identified. *S*. *xilosus*, *S. hominis*, *S. epidermidis*, and *S. cohnii* were isolated from treated wastewater from the COY WWTP, while *S. aureus*, *S. xilosus*, *S. sciuri*, and *S. lentus* were collected from treated wastewater from the ACA WWTP. No staphylococci were detected in the treated wastewater from the CAN WWTP. A total of 16 *Staphylococcus* sp. isolates (8.8%) persisted during WWTP treatments (Figure 2).

Among the 182 isolates, 13 staphylococcal species were confirmed with MALDI-TOF MS^®^ and VITEK 2^®^: *S. cohnii* (n = 41), *S. aureus* (n = 40), *S. sciuri* (n = 26), *S. xylosus* (n = 20), *S*. *saprophyticus* (n = 17), *S. epidermidis* (n = 15), *S. hominis* (n = 9), *S. haemolyticus* (n = 8), *S. simulans* (n = 2), *S. capitis*, *S. lentus*, *S. arlettae*, and *S. warneri* with one isolate each. Only five species (*S. aureus*, *S. conii*, *S. sciuri*, *S. saprophyticus*, and *S. epidermidis*) were present in the four WWTPs studied (Figure 3).

A total of 248 isolates of *Enterococcus* sp. were obtained. The distribution by WWTP was as follows: ACA (n = 58), COY (n = 37), CAN (n = 71), and NUT (n = 82). The number of *Enterococcus* sp. collected in community WWTPs (n = 95) was lower than that obtained from hospital WWTPs (n = 153). The seasons with the highest frequency of *Enterococcus* sp. isolates were autumn and winter (n = 143, 57.7%) (Figure 1).

The analysis by type of water (raw or treated) showed that 185 isolates (74.6%) were collected in the raw wastewater. *E*. *faecium* was present most frequently in treated wastewater from the ACA WWTP and less frequently in the COY and CAN WWTPs. A total of 63 *Enterococcus* sp. isolates (25%) persisted during WWTP treatments (Figure 4).

Seven enterococci species were confirmed with MALDI-TOF MS^®^ and VITEK 2^®^: *E. faecalis* (n = 148), *E*. *faecium* (n = 58), *E*. *hirae* (n = 37), *E*. *casseliflavus* (n = 2), *E*. *avium*, *E*. *durans*, and *E*. *gallinarum* with one isolate each. The distribution of enterococci species by WWTP showed that only three species (*E*. *faecalis*, *E*. *faecium*, and *E*. *hirae*) were present in all the WWTPs studied (Figure 5).

### 3.2. Antimicrobial Susceptibility of Staphylococcus sp. and Enterococcus sp. Isolates

*Staphylococcus* sp. isolates were resistant to oxacillin (n = 71, 39%), erythromycin (n = 62, 34.1%), clindamycin (n = 25, 13.7%), tetracycline (n = 23, 12.6%), daptomycin (n = 15, 8.2%), levofloxacin/moxifloxacin (n = 13 each one, 7.1%), ciprofloxacin (n = 11, 6%), gentamicin (n = 8, 4.4%), trimethoprim/sulfamethoxazole (n = 8, 4.4%), doxycycline (n = 7, 3.8%), linezolid (n = 5, 2.7%), rifampicin (n = 3, 1.6%), vancomycin (n = 2, 1.1%), and tigecycline (n = 5, 2.7%).

The oxacillin-resistant strains showed MIC 1–>4 mg/L and were mainly found in raw water, but also in treated wastewater. This phenotype was found in 10 of the 13 species of staphylococci identified, mainly in *S. cohnii*, *S. sciuri*, *S. saprophyticus*, and *S. epidermidis*. Only one strain of methicillin-resistant *S. aureus* (MRSA) was found in the raw wastewater from the ACA WWTP. Seven staphylococcal isolates collected from raw wastewater, mainly from the WWTP of CAN, showed resistance to doxycycline MIC 8–>16 mg/L. Erythromycin resistance was observed in 62 isolates; 52 had MIC > 8 mg/L; these were isolated mainly from raw and treated wastewater from the WWTPs of ACA, COY, and CAN. Clindamycin-resistant *Staphylococcus* sp. showed MIC > 4 mg/L, mostly detected in the raw wastewater (23/25, 92%) from the WWTPs studied. This phenotype was found in 8 of the 13 staphylococcal species identified, principally *S. cohnii*, *S. epidermidis*, and *S. xilosus*. The presence of *Staphylococcus* sp. resistant to fluoroquinolones MIC 2–>8 mg/L was observed in the raw wastewater from ACA and CAN WWTPs. The species that mainly showed this resistance were *S. epidermidis* and *S. haemolyticus*. Vancomycin resistance was detected in two strains (*S. epidermidis* and *S. lentus*) isolated from raw wastewater from CAN WWTP and treated wastewater from ACA WWTP, respectively; these two strains showed a multi-resistance profile, with MIC values > 8 mg/L for erythromycin, linezolid, and daptomycin, and >32 mg/L for vancomycin. Daptomycin-resistant *S. epidermidis*, *S. xilosus*, *S. lentus*, *S. cohnii*, and *S. sciuri* strains with MIC 2–>8 mg/L were detected mainly in the raw wastewater from ACA, CAN, and NUT. Two strains (*S. lentus* MIC > 8 mg/L and *S. sciuri* MIC 2 mg/L) were detected in the treated wastewater of ACA WWTP. Linezolid-resistant *S. aureus*, *S. epidermidis*, *S. xilosus*, and *S. cohnii* strains were isolated from ACA (n = 3), COY (n = 1), and CAN (n = 1) raw wastewater, and *S. lentus* was collected from ACA-treated wastewater; all strains showed MIC > 8 mg/L. Five strains (*S. conhii*, *S. saprophyticcus*, *S. hominis*, and *S. epidermidis*) isolated from raw wastewater from ACA, COY, and CAN WWTPs were not sensitive to tigecycline with MICs of 0.5–1 mg/L; two of them (*S. conhii* and *S. epidermidis*) were multidrug resistant. Thirty-one *Staphylococcus* sp. isolates with multi-resistance profiles were found in wastewater: CAN (n = 10), ACA (n = 8), COY (n = 8), and NUT (n = 5). Twenty-nine isolates were collected from raw and two from treated wastewater. This multi-resistance was observed in strains of *S. epidermidis* (n = 8), *S. cohnii* (n = 8), *S. hominis* (n = 4), *S. sciuri* (n = 3), *S. xilosus* (n = 3), *S. haemolyticus* (n = 2), *S. arlettae* (n = 1), *S. saprophyticus* (n = 1), and *S. lentus* (n = 1). The wastewater with the lowest number of isolates of *Staphylococcus* sp. resistant to all tested antibiotics came from NUT (Appendix A).

*Enterococcus* sp. isolates were resistant to erythromycin (n = 174, 70.2%), tetracycline (n = 112, 45.2%), doxycycline (n = 103, 41.5%), nitrofurantoin (n = 53, 21.4%), ciprofloxacin (n = 29, 11.7%), levofloxacin (n = 28, 11.3%), daptomycin (n = 27, 10.9%), and ampicillin (n = 6, 2.4%); all isolates were sensible to vancomycin, linezolid, and tigecycline.

More than 70% of the enterococci isolated were resistant to erythromycin, only observed in *E. faecium* and *E. faecalis* strains isolated mainly from hospital wastewater (n = 117). In total, 119 strains with this resistance phenotype were collected from raw wastewater from ACA, COY, CAN, and NUT, and 45 strains from treated wastewater, of which 35/45 were isolated from community wastewater from ACA and COY WWTPs. Resistance to ciprofloxacin (24/29) with MICs of 2–4 mg/L and levofloxacin (23/28) with MICs of 4–8 mg/L was observed predominantly in *E. faecium* strains; almost half of these strains were found in treated wastewater from ACA and CAN WWTPs.

Resistance to daptomycin (MIC 4 mg/L) was only found in *E. faecalis* strains isolated mainly from raw wastewater from ACA, CAN, and NUT WWTPs; only five strains were isolated from treated wastewater (CAN n = 3 and ACA n = 2). The highest number of strains (23/27) with this resistance phenotype were isolated from hospital wastewater (CAN and NUT). Ninety-four *Enterococcus* sp. isolates with multi-resistance profiles were found in wastewater: NUT (n = 33), CAN (n = 26), ACA (n = 26), and COY (n = 9). Sixty-one of these isolates were collected from raw wastewater, and thirty-three from treated wastewater. This multi-resistance was observed in *E. faecalis* (n = 59), *E. faecium* (n = 32), *E. hirae* (n = 2), and *E. gallinarum* (n = 1) (Appendix A).

## 4. Discussion

The present study demonstrated that hospital and community wastewater, both raw and treated, represents a reservoir of staphylococci and enterococci resistant to antibiotics of last choice, as a clearance rate of 46.1% and a survival rate of 19% were detected, making these results important from a public health and environmental standpoint.

No significant differences were observed between the frequency of isolates of staphylococci from community versus hospital WWTPs; this may demonstrate the ubiquitous ability of staphylococci to thrive in wastewater from community and hospital settings, while the frequency of enterococci was higher in hospital WWTPs, probably associated with the nosocomial origin of this genus. The highest frequency of *Staphylococcus* sp. isolates was observed in spring and summer (70.8%), while in the case of *Enterococcus* sp. it was in autumn and winter (57.7%), indicating that staphylococcus populations decreased in the cold season; on the other hand, enterococci remained almost constant throughout the year, although they were more frequent in autumn and winter. The seasonal distribution of some microorganisms has been pointed out in other works, in which several contributing factors are highlighted. Kang et al. reported that the diversity and relative abundance of bacterial communities in wastewater collected in winter were significantly lower compared to summer samples, and that temperature and dissolved oxygen were the main factors driving seasonal changes in bacterial diversity, richness, and community structure in the WWTPs studied [21]. In another work, He Y et al. found that temperature and chemical oxygen demand were the main environmental factors affecting the bacterial community structure of the wastewater tested, with temperature having the greatest effect on species composition [22]. Given this background, it is likely that the seasonal variations observed in the frequency of *Staphylococcus* sp. and *Enterococcus* sp. were maybe influenced by the temperature of the wastewater collected in each season of the year.

In relation to the number of *Staphylococcus* sp. And *Enterococcus* sp. isolates by type of sample (raw or treated), it was observed that both genera were isolated more frequently in the raw wastewater 91.2% and 74.6%, respectively. This is related to the fact that one of the functions of WWTPs is to decrease the bacterial load as it passes through the different treatment systems. Other studies have reported the presence of a higher abundance of bacteria in raw wastewater [15,17,23]. Interestingly, although there are more than 40 different CoNS species described, only some of them were found in treated wastewater, these species have been previously reported to several hospital infections in Mexico [10]. Of all the staphylococci collected, only 8.2% were found in the treated wastewater, the latter being those that survived treatment. Only seven species of staphylococci showed the ability to survive the treatment systems (*S. aureus*, *S. epidermidis*, *S. xilosus*, *S. hominis*, *S. cohnii*, *S. sciuri*, and *S. lentus*) (Figure 2), these results are like those reported by Gomez et al. where they reported 5 strains of *S. aureus* and 14 isolates of CoNS in treated wastewater from 6 WWTPs in Spain [24], while the work by Heβ and Gallert found 16 different CoNS species, predominantly observing the presence of *S. saprophyticus*, *S. sciuri*, *S. xilosus*, *S. lentus*, and *S. cohnii* in treated wastewater from two WWTPs in Germany [25]; these findings are similar to our results.

Of the 13 staphylococcal species detected in this study, 8 were the majority (*S. cohnii*, *S. aureus*, *S. sciuri*, *S. xylosus*, *S. saprophyticus*, *S. epidermidis*, *S. hominis*, and *S. haemolyticus*) (Figure 3). The presence of these species has also been reported in wastewater analyzed from hospital or community WWTPs in Germany, Portugal, Nigeria, and Iran [18,25,26,27], these results being like those reported in this study. Interestingly, these species cause serious infections in humans, mainly *S. aureus*, *S. epidermidis*, *S. hominis*, and *S. saprophyticus* [10,11]. According to CAN and NUT hospital records (from August 2021 to September 2022), these four species ranked first as causing healthcare-associated infections (HAIs) in both hospitals (Microbiologic Laboratories’ data).

Among the *Enterococcus* sp. isolated from treated wastewater, only 25% survived treatment; the species that showed the greatest ability to survive the treatment systems were *E. faecium* (n = 37), *E. faecalis* (n = 12), and *E. hirae* (n = 11) strains (Figure 4). In contrast to these results, Alduhaidhawi reported *E. faecalis* (n = 50) as the most prevalent isolate in wastewater samples, followed by *E. faecium* (n = 35) [28]; while Gotkowska-Płachta, reported that *E. faecium* (42.9%) was the most prevalent isolate in the wastewater studied followed by *E. faecalis* (31.0%), like the result of this study [29]. In a study by Molale-Tom et al. in South Africa, the most prevalent species in the wastewater samples analyzed were *E. hirae* (21.0%), followed by *E. faecalis* (21.0%) and *E. faecium* (19.0%) [30]; these three species were also found in the residual and treated wastewater in this study.

The *Enterococcus* sp. isolates detected in this work were grouped into seven different species, of which three were the most abundant (Figure 5), *E. faecium*, *E. faecalis*, and *E. hirae*, the presence of these three species has also been predominantly detected in hospital and community WWTP wastewater from Poland, Iran, South Africa, England, and Canada [28,29,30,31], these results being like those observed in our study. According to the literature, *E. faecium* and *E. faecalis* are the most frequently isolated species in HAIs, such as meningitis, endocarditis, urinary tract infections, and postoperative infections [32,33]. CAN and NUT hospital records (August 2021 to September 2022) for enterococci show that *E. faecium* and *E. faecalis* had the first places as HAI causative, followed by *E. avium* and *E. gallinarum*. *E. faecalis* is used as an indicator of fecal contamination in environmental waters because of its release through human and animal feces; measurement of these bacteria is used in aquatic environments as quality control. Enterococci can acquire antimicrobial resistance by horizontal gene transfer or by the release of trace antibiotics into the environment through hospital wastewater discharges [34,35,36]. In this study, enterococci isolates were detected in both raw and treated water. The presence of specific species of HAI-causing staphylococci and enterococci in CAN and NUT hospitals and their detection in the environment of the analyzed wastewater could represent an important public health problem, given their potential dissemination in the environment through treated wastewater.

According to the World Health Organization’s “AWaRe” classification, antibiotics are classified into three groups: access, surveillance, and reserve [37]. Results obtained from antimicrobial susceptibility profiles of *Staphylococcus* sp. and *Enterococcus* sp. isolates collected from the wastewater studied showed that some strains were resistant to surveillance antibiotics such as ciprofloxacin, levofloxacin, moxifloxacin, and vancomycin, and to reserve antibiotics such as linezolid, daptomycin, and tigecycline. Although raw wastewater had the highest load of staphylococci and enterococci resistant to surveillance and reserve antibiotics, these resistances were also detected in isolates recovered from treated wastewater (Appendix A). The results showed that in staphylococcal isolates, the percentage of resistance to oxacillin and erythromycin was higher than 60%; resistance to these antibiotics has been detected up to 70% in wastewater systems in other studies [38,39,40]; these results are compatible with our findings. MRSA strains were underrepresented in this study, as only a single isolate of methicillin-resistant *S. aureus* was detected, which contrasts with the results of other works, where the presence of MRSA in wastewater was higher [41,42,43,44]. Quinolone resistance in staphylococcal collected from wastewater in this study was around 7%. Faria et al., reported 5.6% of CoNS resistant to ciprofloxacin in community wastewater from a WWTP in Portugal [26], whereas Nogueira et al. found no resistance to ciprofloxacin in their study [45]. Fluoroquinolones have been reported to be mobile in the aquatic environment due to their hydrophilic characteristics, which would explain their presence in these media [46]. Vancomycin resistance was detected in two strains (*S. epidermidis* and *S. lentus*) with MIC values > 32 mg/L and with a multi-resistance profile; these results differ from those reported by Nogueira et al., who found CoNS isolates sensitive to vancomycin and daptomycin in wastewater [45], while Limayem et al. isolated vancomycin-resistant CoNS in community wastewater [47].

Vancomycin resistance in these two species is relevant considering that vancomycin-resistant *S. epidermidis* is a major cause of serious infections in hospitals [48,49]. This strain was isolated from raw wastewater from the WWTP of the CAN hospital, where, according to the antimicrobial susceptibility profile reports of this hospital, no clinical isolates of vancomycin-resistant staphylococci were detected during the wastewater collection period. Probably, the presence of traces of vancomycin in wastewater and the ability of this species to produce biofilms could have contributed to the selection of the vancomycin-resistant *S. epidermidis* strain. This may be supported by experimental results obtained by Sakimura et al, who found that sensitive biofilm-forming *S. epidermidis* strains express resistance to vancomycin shortly after attachment to a metal surface and the presence of the antibiotic in the medium, observing that resistance increased after attachment as the biofilm formed [50]. Another relevant finding of this work was the presence of vancomycin-resistant *S. lentus* in treated wastewater from the COY community WWTP, which probably suggests that this bacterium had the ability to acquire vancomycin resistance within the WWTP or was able to pass through its treatment systems. *S. lentus* is an opportunistic pathogen present primarily in animals, can be in a variety of hosts and in the environment, and has been reported to cause a variety of infections in humans including endocarditis, meningitis, peritonitis, pyometra, and sinusitis [51,52,53,54]. This species has multiple resistance markers that confer the ability to be resistant to several groups of antibiotics, as evidenced by the resistance phenotype of the strain reported in this study. A study published by Shaker et al. reported 17 *S. lentus* strains resistant to vancomycin, levofloxacin, and erythromycin isolated from hospital settings [55]. Although there are several reports on the presence of vancomycin-resistant enterococci in community and hospital wastewater reporting a frequency of 20–>80% [56,57,58], in the present study, this resistance phenotype was not detected in any of the enterococci strains collected from wastewater.

In this study, resistance to daptomycin was found in some of the staphylococcal (8.2%) and enterococcal (10.9%) isolates studied; this differs from that reported by Nogueira et al, who found no daptomycin-resistant staphylococcal isolates in the wastewater they analyzed [45], as opposed to that reported by Limayem et al, who found daptomycin-resistant staphylococci in wastewater from a community WWTP [47], similar to that observed in this study. In the work published by Li et al., they detected enterococci and staphylococci strains resistant to daptomycin, vancomycin, and linezolid in hospital wastewater [59], phenotypes present in some of the strains analyzed in this study. Resistance to linezolid in clinical isolates has been monitored since 2001 through programs such as the SENTRY Antimicrobial Surveillance Program, the Linezolid Experience and Accurate Determination of Resistance (LEADER) initiative, ZAAPS (Zyvox^®^ Annual Appraisal of Potency and Spectrum), and T.E.S.T. (Tigecycline Evaluation and Surveillance Trial), where they report values between 0.5 and 1.7% of resistance to linezolid in Gram-positive bacteria [60]. In our study, we found 2.7% of staphylococcal strains resistant to linezolid in raw and treated wastewater; this resistance was also reported by Li et al. in staphylococci isolated from hospital wastewater [59]. All enterococci isolated in this study were sensitive to linezolid. These results contrast with those reported by Jahne et al., who found a 37% resistance to linezolid in enterococci strains isolated from wastewater from a livestock farm [61], and by Freitas et al. who isolated two *E. faecalis* strains from wastewater in Tunisia with intermediate resistance to linezolid (MIC 4 mg/L) [62], while the work published by Li et al. reported linezolid-resistant enterococci in hospital wastewater [59]. In this study, five strains not sensitive to tigecycline were detected in community and hospital raw wastewater; two of these strains were multidrug-resistant (*S. conhii* and *S. epidermidis*). Tigecycline is a reserve antibiotic that is only used in severe infections that are difficult to treat, so the presence of these strains in community and hospital raw wastewater is relevant. Szewczyk’s et al. study observed that in the hospital environment, *S. cohnii* acquires antibiotic resistance very quickly, which allows it to remain for a long time in the environment. She also found a great diversity of strains, suggesting its ability to spread. She also mentions that possibly, the flexibility with which this species acquires resistance markers in the hospital environment may favor its ability to spread to the community environment, serving as a bridge to transmit these resistance determinants to other species such as *S. epidermidis*, *S. saprophyticus*, and *S. hominis* [63]. This would imply a potential risk of dissemination of resistance to reserve antibiotics.

The results of the study showed that 17% of *Staphylococcus* sp. and 38% of *Enterococcus* sp. collected from raw and treated wastewater were multidrug-resistant. Twenty-eight percent/n = 35 of the total multidrug resistant strains (n = 125) was found in treated wastewater. In the case of staphylococci, the number of multidrug-resistant strains were similar in community wastewater (n = 16) and hospital wastewater (n = 15), while in multidrug-resistant enterococci, strains were higher in hospital wastewater (n = 59) compared to community wastewater (n = 35). These findings are essential, considering that multidrug-resistant bacteria are associated with high mortality rates, high costs of care, and dissemination of resistant bacteria, among other problems. In addition, the species identified in this study that carry this multidrug resistance cause HAIs in hospitals.

## 5. Conclusions

The highest number of multidrug-resistant strains were detected in hospital wastewater, which would indicate that this wastewater, due to its origin, could be contributing the highest load of multidrug-resistant bacteria to the environment. A portion of the treated community and hospital wastewater analyzed in this study is reused to irrigate the green areas of the city and the hospital, respectively; the remainder is discharged to the municipal sewer. It has been documented in the literature that the three types of treatment used in these WWTPs perform their function by reducing the number of bacteria present in the treated wastewater, even when some of these bacteria manage to survive the treatment. The presence of multidrug-resistant bacteria in treated wastewater can have an ecological impact by spreading to the environment. The study of a larger number of WWTPs in different regions of our country would allow us to have more evidence of the frequency of multidrug-resistant strains in treated wastewater, directing efforts towards regulations governing the discharge of these bacteria into the environment, thus minimizing the risk of exposure to these pathogens. Finally, the development of new wastewater treatment technologies could emerge as a necessity to combat the problem of AMR in these environments.

## Figures and Tables

**Figure 1 microorganisms-12-00645-f001:**
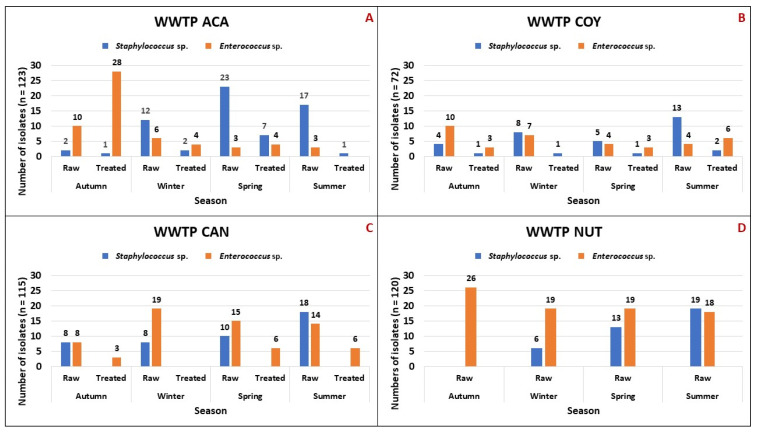
Seasonal variation in the abundance of *Staphylococcus* sp. and *Enterococcus* sp. isolates in raw and treated wastewater from community and hospital WWTPs. (**A**) Community Wastewater Treatment Plant “Acapantzingo” (WWTP ACA) in Cuernavaca City. (**B**) Community Wastewater Treatment Plant “Coyoacán” (WWTP COY) in Mexico City. (**C**) Hospital Wastewater Treatment Plant “Cancerología” (WWTP CAN) in Mexico City. (**D**) Hospital Wastewater Treatment Plant “Nutrición” (WWTP NUT) in Mexico City.

**Figure 2 microorganisms-12-00645-f002:**
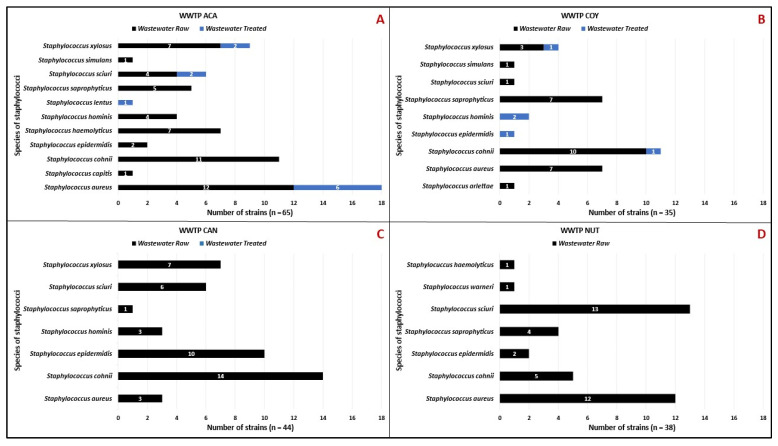
Distribution of staphylococcal species by type of wastewater (raw/treated) in the WWTPs analyzed. (**A**) Community Wastewater Treatment Plant “Acapantzingo” (WWTP ACA) in Cuernavaca City. (**B**) Community Wastewater Treatment Plant “Coyoacán” (WWTP COY) in Mexico City. (**C**) Hospital Wastewater Treatment Plant “Cancerología” (WWTP CAN) in Mexico City. (**D**) Hospital Wastewater Treatment Plant “Nutrición” (WWTP NUT) in Mexico City.

**Figure 3 microorganisms-12-00645-f003:**
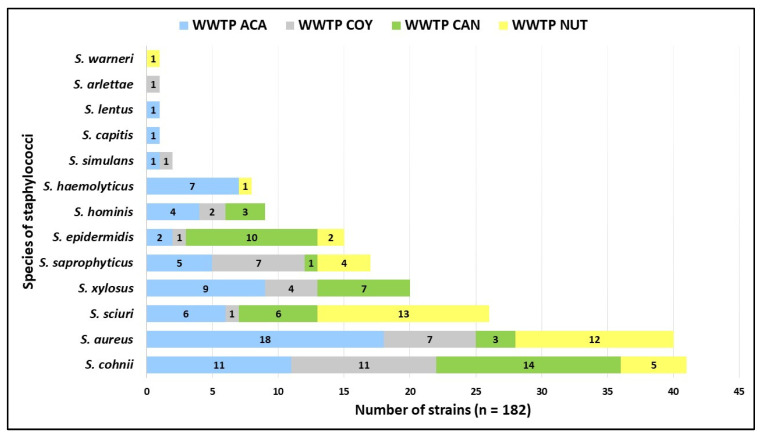
Distribution of staphylococcal species by WWTP analyzed. Community Wastewater Treatment Plant “Acapantzingo” (WWTP ACA) in Cuernavaca City. Community Wastewater Treatment Plant “Coyoacán” (WWTP COY) in Mexico City. Hospital Wastewater Treatment Plant “Cancerología” (WWTP CAN) in Mexico City. Hospital Wastewater Treatment Plant “Nutrición” (WWTP NUT) in Mexico City.

**Figure 4 microorganisms-12-00645-f004:**
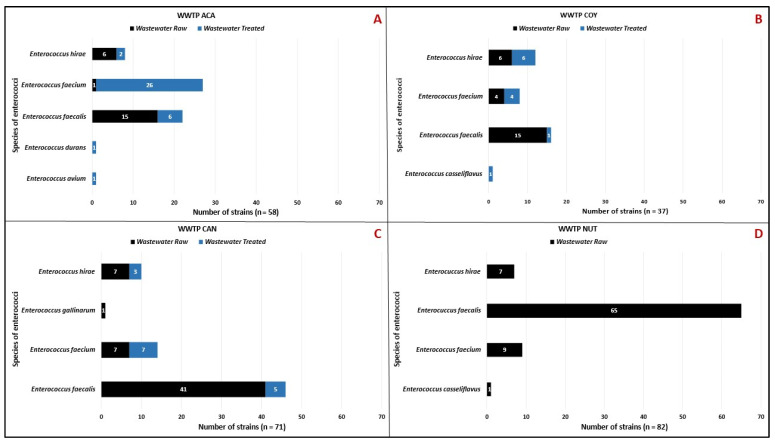
Distribution of enterococci species by type of wastewater (raw/treated) in the WWTPs analyzed. (**A**) Community Wastewater Treatment Plant “Acapantzingo” (WWTP ACA) in Cuernavaca City. (**B**) Community Wastewater Treatment Plant “Coyoacán” (WWTP COY) in Mexico City. (**C**) Hospital Wastewater Treatment Plant “Cancerología” (WWTP CAN) in Mexico City. (**D**) Hospital Wastewater Treatment Plant “Nutrición” (WWTP NUT) in Mexico City.

**Figure 5 microorganisms-12-00645-f005:**
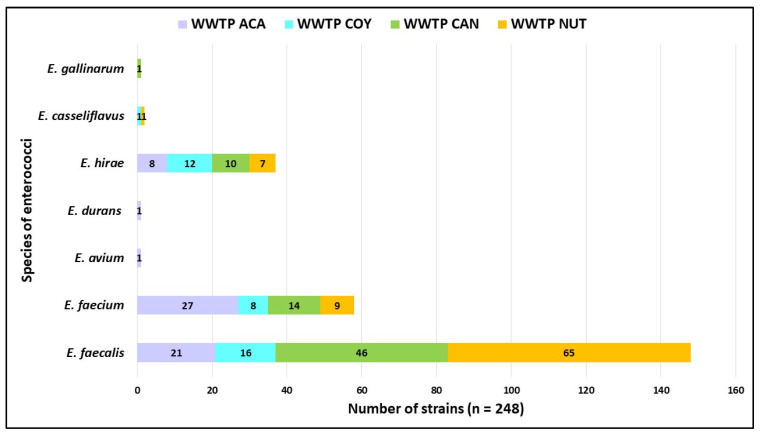
Distribution of enterococcus species by WWTP analyzed. Community Wastewater Treatment Plant “Acapantzingo” (WWTP ACA) in Cuernavaca City. Community Wastewater Treatment Plant “Coyoacán” (WWTP COY) in Mexico City. Hospital Wastewater Treatment Plant “Cancerología” (WWTP CAN) in Mexico City. Hospital Wastewater Treatment Plant “Nutrición” (WWTP NUT) in Mexico City.

## Data Availability

Data are contained within the article.

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
