# Peer review of "Multidrug-Resistant Staphylococcus sp. and Enterococcus sp. in Municipal and Hospital Wastewater: A Longitudinal Study"

_microorganisms, 2024, doi:10.3390/microorganisms12040645_

Round 1

Reviewer 1 Report

Comments and Suggestions for Authors

Velazquez-Meza et al. have isolated Staphylococcus sp and Enterococcus sp from wastewater influent and effluent and evaluated their AMR susceptibility. The study was designed and concluded well, with no issue in its writing. They have done an excellent job. Still, I do not see anything in Figures 6 and 7 except green-red decoration. I assume heatmaps should have the scale and color description, that is lacking here. The current format of figures 6-7 should not be published.  Next, Enterococcus sp are relatively host-specific (https://doi.org/10.2166/wh.2018.293), please mention that on discussion, and interpreted pathogenic Enterococcus sp could be from human infection sources but possible environmental species of Enterococcus sp could acquire AMR capacity from gene transfer or acquired from the environmental release of antimicrobial agents or these were natural. Yes, some species of Enterococcus are used as FIB for determining microbial contamination (https://doi.org/10.3390/ijerph18115513). Also, monitoring ARB in wastewater can be used to explain pathogens circulating in the community and assess human health risks (https://doi.org/10.1016/j.envres.2023.118052). Overall, the study has done well and written well. The paper will add value to the scientific community and advance the field.  

Author Response

I am very grateful for the reviewers' comments, as they contribute to enrich the work. Below we give a timely response to each of them.

Reviewer 1.

Commentary: Velazquez-Meza et al. have isolated Staphylococcus sp and Enterococcus sp from wastewater influent and effluent and evaluated their AMR susceptibility. The study was designed and concluded well, with no issue in its writing. They have done an excellent job. Still, I do not see anything in Figures 6 and 7 except green-red decoration. I assume heatmaps should have the scale and color description, that is lacking here. The current format of figures 6-7 should not be published.  Next, Enterococcus sp are relatively host-specific (https://doi.org/10.2166/wh.2018.293), please mention that on discussion, and interpreted pathogenic Enterococcus sp could be from human infection sources but possible environmental species of Enterococcus sp could acquire AMR capacity from gene transfer or acquired from the environmental release of antimicrobial agents or these were natural. Yes, some species of Enterococcus are used as FIB for determining microbial contamination (https://doi.org/10.3390/ijerph18115513). Also, monitoring ARB in wastewater can be used to explain pathogens circulating in the community and assess human health risks (https://doi.org/10.1016/j.envres.2023.118052). Overall, the study has done well and written well. The paper will add value to the scientific community and advance the field.  

Answer: Figures 6 and 7 were separated by PTAR to enlarge the image and are included in the supplementary material.

Answer: In accordance with the reviewer's suggestions, the following paragraph is included in the discussion:

  1. faecalis is used as an indicator of fecal contamination in environmental waters because of its release through human and animal feces; measurement of these bacteria is used in aquatic environments as quality control. Enterococci can acquire antimicrobial resistance by horizontal gene transfer or by the release of trace antibiotics into the environment through hospital wastewater discharges. In this study, enterococci isolates were detected in both raw and treated water. Discussion section. LINES 338-343

Reviewer 2 Report

Comments and Suggestions for Authors

Dear Authors,

please revise the manuscript or answer the following questions.

I propose change a title for “Multi-drug resistant Staphylococcus sp. and Enterococcus sp. in municipal and hospital wastewater – a case study of Mexico City and Cuernavaca City.”

The Authors describe first municipal wastewater, and second hospital wastewater. Why there are no data of treated wastewater from hospital with a single sump (NUT)? How many multi-drug resistant bacteria was still in treated wastewater?

The methodology lacks information about primary, secondary, and tertiary treatment systems every four WWTPs. There are the same treatment processes and installations?

Different suspected staphylococcal and enterococcal colonies of bacteria were selected and identified by mass spectrometry – these colonies were from treated wastewater?

The conclusions are too concise. Please also add information about your research plans related to environmental threats. Are there any processes or installations that would remove these bacteria?

Figure 1 – please give the same scale for all WWTPs.

Figure 3 - 13 types of bacteria have been identified and only 7 are visible.

Figure 4 – please give the same scale for all WWTPs.

Author Response

I am very grateful for the reviewers' comments, as they contribute to enrich the work. Below we give a timely response to each of them.

Reviewer 2

Commentary: I propose change a title for “Multi-drug resistant Staphylococcus sp. and Enterococcus sp. in municipal and hospital wastewater – a case study of Mexico City and Cuernavaca City.”

Answer: The title was modified considering most of the suggestions of the reviewer, who preferred to add the type of study instead of the location. LINES 2, 3

Multidrug-resistant Staphylococcus sp and Enterococcus sp in municipal and hospital wastewater: Longitudinal study.

Question: Why there are no data of treated wastewater from hospital with a single sump (NUT)?

Answer: In the NUT hospital we do not have data on treated wastewater, because in that hospital the wastewater does not go through primary, secondary and tertiary treatment, there is only a pretreatment system that separates large waste and from there the water goes directly to the municipal sewer. For this reason, there is no sample of treated wastewater.

Question: How many multi-drug resistant bacteria was still in treated wastewater?

Answer: Staphylococcus sp two and Enterococcus sp thirty-three

These data are described in the results section, LINES 240-241 263-264

Twenty-nine isolates were collected from raw and two from treated wastewater.

Sixty-one of these isolates were collected from raw wastewater, and thirty-three from treated wastewater.

Question: The methodology lacks information about primary, secondary, and tertiary treatment systems every four WWTPs. There are the same treatment processes and installations.?

Answer: We address this observation by including information on the treatment systems of the WWTPs studied in the methodology section. LINES 105-109

“The layout of both WWTPs begins with a primary treatment, then the water flow is dosed to an aeration process mediated by regulating valves in different chambers. The treatment process continues with a secondary phase of clarification and composite sedimentation, precipitating the sludge. The tertiary treatment includes calcium hypochlorite tablets, granular activated carbon filters and ultraviolet light (UVL) in the effluent.”

Question: There are the same treatment processes and installations.?

Answer: Yes, they are

Question: Different suspected staphylococcal and enterococcal colonies of bacteria were selected and identified by mass spectrometry – these colonies were from treated wastewater?

Answer: Based on your comment, the paragraph has been reworded to make the information clearer.

“The different suspected staphylococcal and enterococcal colonies isolated from the raw wastewater and treated wastewater were identified first by chromoagar and subsequently identified by mass spectrometry, using the Matrix-Assisted Laser Desorption/Ionization-Time of Flight mass spectrometry (MALDI-TOF MS®) (Bruker Daltonics, Bremen, Germany) equipment. Methodology section. LINES 119-123

These colonies were from treated wastewater.

Raw and treated wastewater

Question: The conclusions are too concise. Please also add information about your research plans related to environmental threats. Are there any processes or installations that would remove these bacteria?

Based on your comment the conclusion is modified. Conclusion section. LINES  441-456

Answer: The highest number of multidrug-resistant strains was detected in hospital wastewater, which would indicate that this water, due to its origin, could be contributing the highest load of multidrug-resistant bacteria to the environment.  A portion of the treated community and hospital wastewater analyzed in this study is reused to irrigate the green areas of the city and the hospital, respectively; the remainder is discharged to the municipal sewer. It has been documented in the literature that the three types of treatment used in these WWTPs perform their function by reducing the number of bacteria present in the treated wastewater, even when some of these bacteria manage to survive the treatment. The presence of Mult resistant bacteria in treated water can have an ecological impact by spreading to the environment. The study of a larger number of WWTPs in different regions of our country would allow us to have more evidence of the frequency of multidrug-resistant strains in treated wastewater, directing efforts towards regulations governing the discharge of these bacteria into the environment, thus minimizing the risk of exposure to these pathogens. Finally, the development of new wastewater treatment technologies could emerge as a necessity to combat the problem of AMR in these environments.

Commentary: Figure 1 – please give the same scale for all WWTPs.

Answer: Based on your comment the images were adjusted to the same scale, thank you.

Commentary: Figure 3 - 13 types of bacteria have been identified and only 7 are visible.

Answer: Based on your comment, the correction has been made.

Commentary: Figure 4 – please give the same scale for all WWTPs.

Answer: Based on your comment the images were adjusted to the same scale.

Reviewer 3 Report

Comments and Suggestions for Authors

There are still some problems in this manuscript. I don't recommend this manuscript to be published in this form. English could be improved. There are some sentences with grammar mistakes and missing articles throughout the manuscript. Some sentences need to be reworded. Figures are not clear. The authors failed to format fonts, sentences  in a scientific manner. The author should pay more attention to the writing details.

 The study is preliminary and does not conclude any solid conclusion from these results.

The motive of this study is not clear and results made from the experiments are not satisfactory

Comments on the Quality of English Language

There are some sentences with grammar mistakes and missing articles throughout the manuscript.

Author Response

I am very grateful for the reviewers' comments, as they contribute to enrich the work. Below we give a timely response to each of them.

Reviewer 3

  1. There are still some problems in this manuscript. I don't recommend this manuscript to be published in this form. English could be improved. There are some sentences with grammar mistakes and missing articles throughout the manuscript. Some sentences need to be reworded. Figures are not clear. The authors failed to format fonts, sentences in a scientific manner. The author should pay more attention to the writing details. The study is preliminary and does not conclude any solid conclusion from these results.

  1. The motive of this study is not clear, and results made from the experiments are not satisfactory

Answer: Considering the comments of the other reviewers, the suggested changes were made.

Reviewer 4 Report

Comments and Suggestions for Authors

Dear Authors,

The presented study is interesting and provides important new data. However, before it can be published, the manuscript needs a substantial improvement.

Below you can find my remarks:

First of all, the title is too general. The fact that hospital wastewater is or may be reservoir of multidrug bacteria is known. I suggest that the title should be changed to introduce the actual scope of the study and the fact that elimination of these bacteria through WW treatment was assessed.

Abstract, l.22-23: it is not presumptive isolation of staphylococci and enterococci, but isolation of presumptive staphylococci and enterococci.

l. 25-26: information about the number of WW samples examined should be moved upwards, after the sentence about WW sample collection.

l. 33: Abstract should be short, but informative. “Some strains” does not mean anything. Please provide the number or percentage.

Keywords: only bacterial names should be written in italics.

Introduction is too scarce. It should contain information about multidrug resistance – its prevalence and consequences. It should also refer to treatment plant elimination of bacteria and AMR bacteria. What is already known? Where is the actual knowledge gap that you try to fill with your research?

The aim of the study is also vague. The way that it is specified now suggests that the study is of broad context while the Authors focus on two locations with three sites. I understand that the location of the study was probably defined by the affiliation of the Authors, but are there any other reasons for selection of Mexico City apart from that? I can see that the selection of the specific sites was justified in the Materials and Methods section, but I mean the general location (I assume that e.g. Mexico City is inhabited by large numbers of people etc., perhaps there are some problems with purity of water or else, that would be a good justification of the study being conducted where it was?).

Materials and Methods – Antimicrobial Susceptibility Testing: please specify the number of strains that were subjected to the tests. I mean not only the total number of strains, but individual numbers of: staphylococci, enterococci with division into the sites they were derived from.

Results:

Figure 1: I suggest to add explanation of abbreviations into the figure caption, even though they were explained in the Materials and Methods. It will make the figures easier to be understood while using them as stand-alone information. Also, add a total number of isolates obtained from each WWTP, either in the figure caption or by the y axis.

Figure 2 is illegible in this layout. It should be definitely enlarged.

Figure 4: if you use lettering by the figures, you should then explain their meaning in the figure caption.

Figure 6 and 7 are completely non-informative and illegible. Information provided in these figure MUST be presented otherwise, especially that they show the most important outcomes of this study.

Even though the multidrug resistance is mentioned in the manuscript title, there are like two sentences describing this phenomenon in this manuscript. Either change the general aim of the study or pay more attention to the MDR in your results.

Discussion

l. 226-228: This fragment should be rewritten. If such information is important from public health and environmental standpoint (and it is, I do not argue with that), the Authors should here provide more complex view. Do they refer to raw or treated wastewater? If raw WW – what implications does this information have? If treated – the same. If both – you should refer to the removal rate and the survival rate.

l. 229-248: what type of wastewater do the Authors discuss here?

The paragraph that describes the prevalence of staphylococcal species in raw and treated wastewater should also refer to their survival rate from raw to treated WW. In the present version of the discussion, we can only see what species were isolated from treated WW. It is important whether there are species that are more likely to survive the treatment process. This is actually the most important outcome of this study.

The same remark refers to the paragraph describing the enterococcal species.

l. 325: why do the Authors assume that vancomycin is present in wastewater? Do they have such information?

l. 379-380: As I stated before – the multidrug resistance of bacterial isolates is one of the most important outcomes here. This sentence is very general, it is not clear how many of strains (and percentage) isolated from treated wastewater were multidrug resistant. Environmentally this is crucial information.

Conclusions are currently rather a brief summary of the outcomes of the experiments. I suggest to rewrite this paragraph. First summarize the results, but not as briefly as it is now. Then, try to elucidate what are the implications of the presented research results? Are there any future directions that could be suggested here?

Comments on the Quality of English Language

The English language used is generally fine. There are very minor issues (e.g. spelling), that can be improved even during the final preparation of the manuscript (proof).

Author Response

I am very grateful for the reviewers' comments, as they contribute to enrich the work. Below we give a timely response to each of them.

Reviewer 4

The presented study is interesting and provides important new data. However, before it can be published, the manuscript needs a substantial improvement.

Below you can find my remarks:

Commentary: First of all, the title is too general. The fact that hospital wastewater is or may be reservoir of multidrug bacteria is known. I suggest that the title should be changed to introduce the actual scope of the study and the fact that elimination of these bacteria through WW treatment was assessed.

Answer: Based on your comment, the modification has been made. Title section. LINES 2,3

Multidrug-resistant Staphylococcus sp and Enterococcus sp in municipal and hospital wastewater: Longitudinal study.

Commentary: Abstract, l.22-23: it is not presumptive isolation of staphylococci and enterococci, but isolation of presumptive staphylococci and enterococci.

Answer: Based on your comment, the modification has been made.  Abstract section LINES 24,25.

“Screening and presumptive identification of staphylococci and enterococci was performed using chromoagars”. 

Commentary: l. 25-26: information about the number of WW samples examined should be moved upwards, after the sentence about WW sample collection.

Answer: It is described on this site because it is part of the results.

Commentary: l. 33: Abstract should be short, but informative. “Some strains” does not mean anything. Please provide the number or percentage.

Answer: Based on your comment, the information is included in the Abstract section. LINES 34-36

Answer: Resistance to vancomycin (1.1%), linezolid (2.7%) and daptomycin (8.2%/10.9%%) and lower susceptibility to tigecycline (2.7%) were observed.

Commentary: Keywords: only bacterial names should be written in italics

Answer: Based on your comment, the correction was made. Keywords section. LINE 39

Keywords: Wastewater, Multi resistance, Persistence, Staphylococcus sp, Enterococcus sp.

Commentary: Introduction is too scarce. It should contain information about multidrug resistance – its prevalence and consequences. It should also refer to treatment plant elimination of bacteria and AMR bacteria. What is already known? Where is the actual knowledge gap that you try to fill with your research?

Answer: Based on your comment, the modification has been made. Introduction section. LINES  46-57 LINES 79-83

Commentary and Question: The aim of the study is also vague. The way that it is specified now suggests that the study is of broad context while the Authors focus on two locations with three sites. I understand that the location of the study was probably defined by the affiliation of the Authors, but are there any other reasons for selection of Mexico City apart from that? I can see that the selection of the specific sites was justified in the Materials and Methods section, but I mean the general location (I assume that e.g. Mexico City is inhabited by large numbers of people etc., perhaps there are some problems with purity of water or else, that would be a good justification of the study being conducted where it was?).

Answer: Considering your comments the objective has been rewritten Abstract, Objective section LINES 18-20 LINES 83-86

“The objective of the study was to detect multidrug-resistant Staphylococcus sp and Enterococcus sp strains in municipal and hospital wastewater and to determine their elimination or persistence after wastewater treatment.

Answer: Based on your comments, the following information has been added. Methodology section. LINES 100-101

The hospital WWTPs included in this study were selected because they are tertiary-level hospitals that serve a large population in Mexico City and surrounding cities, as well as for their location and ease of sampling. Mexico City has 22,281,442 inhabitants, making it one of the most populated cities in Mexico.

Commentary: Materials and Methods – Antimicrobial Susceptibility Testing: please specify the number of strains that were subjected to the tests. I mean not only the total number of strains, but individual numbers of: staphylococci, enterococci with division into the sites they were derived from.

Answer: Considering your comments the requested data are included. Methodology section. LINES 125-127

The antimicrobial susceptibility profiles of Staphylococcus sp (n=182) [100 community wastewater/82 Hospital wastewater] and Enterococcus sp (n=248) [95 community wastewater/153 Hospital wastewater] isolates were performed by the VITEK 2® automated system (BioMérieux, Marcy l’Etoile, France).

Commentary: Figure 1: I suggest to add explanation of abbreviations into the figure caption, even though they were explained in the Materials and Methods. It will make the figures easier to be understood while using them as stand-alone information. Also, add a total number of isolates obtained from each WWTP, either in the figure caption or by the y axis.

Answer: Considering your comments the recommendations of all the figures were addressed.

Commentary: Figure 2 is illegible in this layout. It should be definitely enlarged.

Answer: Considering your comments font size has been increased

Commentary: Figure 4: if you use lettering by the figures, you should then explain their meaning in the figure caption.

Answer: Considering your comments the information is included in the figure caption.

Commentary: Figure 6 and 7 are completely non-informative and illegible. Information provided in these figure MUST be presented otherwise, especially that they show the most important outcomes of this study

Answer: Considering your comments, Figures 6 and 7 were separated by PTAR to enlarge the image and are included in the supplementary material.

Commentary: Even though the multidrug resistance is mentioned in the manuscript title, there are like two sentences describing this phenomenon in this manuscript. Either change the general aim of the study or pay more attention to the MDR in your results.

Answer: This point was addressed by including information in the introduction. Introduction section.  LINES  46-57

Discussion

Commentary and Question: l. 226-228: This fragment should be rewritten. If such information is important from public health and environmental standpoint (and it is, I do not argue with that), the Authors should here provide more complex view. Do they refer to raw or treated wastewater? If raw WW – what implications does this information have? If treated – the same. If both – you should refer to the removal rate and the survival rate.

Answer: Considering your comments the fragment has been rewritten. Discussion section. LINES  268-271

The present study demonstrated that hospital and community wastewater, both raw and treated, represents a reservoir of staphylococci and enterococci resistant to antibiotics of last choice, as a clearance rate of 46.1% and a survival rate of 19% were detected, making these results important from a public health and environmental point of view.

Question: l. 229-248: what type of wastewater do the Authors discuss here?

“No significant differences were observed between the frequency of isolates of staphylococci from community versus hospital WWTPs;…

Answer: The sentence talks about the frequency of isolates depending on the type of WWTP (community and hospital) both types of water are included.

Commentary: The paragraph that describes the prevalence of staphylococcal species in raw and treated wastewater should also refer to their survival rate from raw to treated WW. In the present version of the discussion, we can only see what species were isolated from treated WW. It is important whether there are species that are more likely to survive the treatment process. This is actually the most important outcome of this study.

Answer: Considering your comments the fragment has been rewritten. Discussion section. LINES  299-303

Of all the staphylococci collected only 8.2% were found in the treated wastewater, the latter being those that survived treatment. Only seven species of staphylococci showed the ability to survive the treatment systems (S. aureus, S. epidermidis, S. xilosus, S. hominis, S. cohnii, S. sciuri and S. lentus) (Figure 2).

Commentary: The same remark refers to the paragraph describing the enterococcal species.

Answer: Considering your comments the fragment has been rewritten. Discussion section. LINES  319-321

Among the Enterococcus sp isolated from treated wastewater, only 25% survived treatment; the species that showed the greatest ability to survive the treatment systems were E. faecium (n=37), E. faecalis (n=12), and E. hirae (n=11) strains; (Figure 4).

Question. 325: why do the Authors assume that vancomycin is present in wastewater? Do they have such information?

Answer: Since we do not have this information, we use the word probably, because it has been documented in the literature that traces of different antimicrobials are present in wastewater.

Commentary: l. 379-380: As I stated before – the multidrug resistance of bacterial isolates is one of the most important outcomes here. This sentence is very general, it is not clear how many of strains (and percentage) isolated from treated wastewater were multidrug resistant. Environmentally this is crucial information.

Answer: Considering your comments, we have included the information. Discussion section. LINES  430-432

The results of the study showed that 17% of Staphylococcus sp and 38% of Enterococcus sp collected from raw and treated wastewater were multidrug resistant; twenty-eight percent/n=35 of the total multidrug resistant strains (n=125) was found in treated water……

Commentary: Conclusions are currently rather a brief summary of the outcomes of the experiments. I suggest to rewrite this paragraph. First summarize the results, but not as briefly as it is now. Then, try to elucidate what are the implications of the presented research results? Are there any future directions that could be suggested here?

Answer: Based on your comment the conclusion is modified. Conclusion section. LINES 441-446

The highest number of multidrug-resistant strains was detected in hospital wastewater, which would indicate that this wastewater, due to its origin, could be contributing the highest load of multidrug-resistant bacteria to the environment.  A portion of the treated community and hospital wastewater analyzed in this study is reused to irrigate the green areas of the city and the hospital, respectively; the remainder is discharged to the municipal sewer. It has been documented in the literature that the three types of treatment used in these WWTPs perform their function by reducing the number of bacteria present in the treated wastewater, even when some of these bacteria manage to survive the treatment. The presence of multidrug-resistant bacteria in treated wastewater can have an ecological impact by spreading to the environment. The study of a larger number of WWTPs in different regions of our country would allow us to have more evidence of the frequency of multidrug-resistant strains in treated wastewater, directing efforts towards regulations governing the discharge of these bacteria into the environment, thus minimizing the risk of exposure to these pathogens. Finally, the development of new wastewater treatment technologies could emerge as a necessity to combat the problem of AMR in these environments.

Round 2

Reviewer 3 Report

Comments and Suggestions for Authors

Author response is not satisfactory and the manuscript does not meet the journal standard. 

Reviewer 4 Report

Comments and Suggestions for Authors

Dear Authors,

Thank you for correcting the manuscript according to the remarks.

It looks much better now.

I recommend its acceptance.